# Two Open Solutions for Industrial Robot Control: The Case of PUMA 560

**Dejan Jokić** [1], **Slobodan Lubura** [2], **Vladimir Rajs** [3,*], **Milan Bodić** [3] and **Harun Šiljak** [4]

1 Department of Electrical and Electronics Engineering, International Burch University, 71000 Sarajevo, Bosnia and Herzegovina; dejan.jokic@ibu.edu.ba
2 Faculty of Electrical Engineering, University of East Sarajevo, 71123 East Sarajevo, Bosnia and Herzegovina; slobodan.lubura@etf.ues.rs.ba
3 Department of Power, Electronic and Telecommunication Engineering, Faculty of Technical Sciences, University of Novi Sad, 21000 Novi Sad, Serbia; milanbodic@uns.ac.rs
4 Connect Centre, Trinity College Dublin, Dublin, Ireland; harun.siljak@tcd.ie
* Correspondence: vladimir@uns.ac.rs; Tel.: +381-638687606

**Abstract:** In this paper we present two different, software and reconfigurable hardware, open architecture approaches to the PUMA 560 robot controller implementation, fully document them and provide the full design specification, software code and hardware description. Such solutions are necessary in today's robotics and industry: deprecated old control units render robotic installations useless and allow no upgrades, advancements, or innovation in an inherently innovative ecosystem. For the sake of simplicity, just the first robot axis is considered. The first approach described is a PC solution with data acquisition I/O board (Humusoft MF634). This board is supported with Matlab Real-Time Windows Toolbox for real-time applications and thus whole controller was designed in Matlab environment. The second approach is a robot controller developed on field programmable gate arrays (FPGA) board. The complexity of FPGA design can be overcome by using a third party software package, such as self-developed Matlab FPGA Real Time Toolbox. In both cases, parameters of motion controller are calculated by using simulation of the PUMA 560 robot first axis motion. Simulations were conducted in Matlab/Simulink using Robotics Toolbox.

**Keywords:** educational robots; MATLAB; robot control; robot programming; open platforms

## 1. Introduction

The PUMA 560 robot made significant impact in the robotics era, and has been widely accepted in many fields of industry. While more advanced robots found their application in industry in recent times, PUMA 560 found its new purpose in education, partially due to the fact that it is the mathematically best described robot. Its simple structure enables development of new controllers and testing of the new controlling algorithms for education and scientific purposes. Nowadays there are many manufacturers in the market, but the produced robots use controllers which are not open for research and education purposes. In education process organized for students it is important to have the possibility to measure different values (position, error, torque etc.) from control algorithms utilized on controller in real time and compare them with results from other simulations as well as textbooks. Consequently, new controlling approaches, as well as controllers for the PUMA 560 robot have been developed at institutes and universities worldwide.

Open architecture community in robotics is not just an educational exercise. It is an immediate necessity, as closed-source solutions harm the repair process, adjustments to the field application, as well as regular maintenance and/or usability after the manufacturers go out of business or change

ownership/business model. In this light, the ability to provide open source software, open source reconfigurable hardware such as field programmable gate arrays (FPGA), and open design for additional adaptor circuitry is instrumental. We need schemes that are modular, reconfigurable, and flexible—they can be re-used for similar robots to those they are originally designed for, their components can be used interchangeably, and mission-specific controllers can be produced by adapting the general architecture. Contribution of this paper is delivering such a solution, with full verification of its performance, and making it available under Creative Commons CC BY 4.0 license.

The motivation we had comes both from the industrial and educational practice, and we contribute to both. When the practitioners ask "Can you repair what you own?" [1], the answer promoted by the majority of equipment vendors is no, and their solutions lack modularity (cannot replace just a part), openness (cannot diagnose faults or replicate the functionality), or flexibility (narrow, restricted set of functions and operations span). We introduce solutions that promote the opposite. In the educational arena [2], open source solutions allow more than closed-source ones: all educators had the experience of closed-source solution providers delivering separate software or hardware units for every functionality needed in the classroom, even if a single solution could cover them all. For example, even if all hardware needed for cascade control of a plant exists in the educational system provided by the vendor, the educators still have to buy a simple (LabView wrapped) application to access the hardware and pay a significant price for it. At the same time, students lose the opportunity to make such an application themselves, or see how it is made, or tweak it as a part of their training, which, we argue is a basic skill for an engineer. Again, here we offer solutions open for editing and tinkering. An open-source trained student will become an open-source advocating professional.

In the majority of scientific papers, one may find three different approaches implemented on different platforms. One of them is pure simulation using MATLAB/Simulink with adequate robotic toolboxes. The second one is using PC-based controllers and the third one is using embedded systems. The first approach, widely accepted and globally in use for education and research, is Robotics Toolbox developed by professor Peter Corke [3]. Major characteristic feature of the mentioned toolbox is the fact it can be used for many other robots, not only PUMA 560. The second approach is very popular because, using a friendly environment on a PC with an appropriate additional data acquisition board, it is possible to control a robot in real time [4–9]. The third approach is also very common due to possibility to develop a standalone embedded controller [10–15]. In Reference [10], an FPGA-based controller for the Fanuc S420F robot is proposed, developed and described as open architecture which enables scalability (possibility of adding new degree of freedom (DOF)) independently from vendors in case of possible failure of old robot where vendors fail to provide further support and maintenance. Comparing a PC-based controller with an embedded controller, it is obvious that different approaches have some advantages and disadvantages, which will be further discussed in our paper.

The inspiration for this work, namely for the new computer-based control results presented, comes from the educational/engineering reality—availability of components and knowledge dictates the strategy of implementation. Previously, we designed a scheme based on an FPGA: when a new acquisition board became available, we decided to examine how both the process of controller implementation and its results compare to the FPGA solution. At the same time, we wanted to deliver both to the community as open source options to choose and use either when hardware resources and application constraints allow it. Since we mostly use the acquisition board as an elaborate AD/DA converter, our solution is not platform-dependent. With that, we avoid the pitfall of relying to yet another proprietary component—the goal of this work is to converge to a completely open and accessible hardware and software scheme.

In the relevant literature there is wide choice of different control strategies with widespread groups of algorithms used for the purpose of controlling the axes of the PUMA 560 robot. For developing a control strategy, it is necessary to use adequate algorithm, as well as parameters in order to provide appropriate controlling signals in real time. Complexity of robot control algorithms very often lies in compromising between accuracy and available hardware resources for the purpose of calculating

torque for particular axis of robot in real time. Due to construction of the PUMA 560 robot and its low accuracy it is unnecessary to use advanced, nonlinear control algorithms. To perform a comparison of results obtained from different platforms it is crucial to use the simplest possible control algorithm which can also be implemented on different platforms. Based on the aforementioned requirements, the plus derivative (PD) algorithm with gravity compensation for robot motion control proposed in [15] is adequate and it represents a compromise between available resources and maximum accuracy which can be achieved with robots like the PUMA 560.

Calculating PD parameters for the algorithm is common practice in simulations, but it is highly demanding in case of controlling a robot axis in real time. Regarding this issue, engineers refer to un-modeled dynamics and unknown friction between some mechanical parts and often propose the use of tuning PD parameters in real time in the course of the control process.

The PUMA 560 robot belongs to the group of anthropomorphic arms. The base configuration corresponds to a two-link planar arm with an additional rotation about an axis of the plane. Axes of base configuration are powered by DC motors (300 W). Remaining three axes correspond to spherical wrist and they are also powered by DC motors (160 W). For testing a modified PD algorithm, the first robot axis is used, since only the first axis of the PUMA 560 robot is not affected by gravity. For gravity compensation it is necessary to use complex control algorithms [15] which can lead to occurrence of discrepancy between results obtained from other platforms and to avoid it the decision was reached to use only the first axis of the PUMA 560 robot for testing purposes.

In the following sections, we present the simulation framework for the first axis of the robot, the hardware and software we developed for the control of the actual robot, and the results of simulation compared to the results of the two controllers on the actual device. The software code and the hardware description is provided in supplementary content for reproducibility and free use.

## 2. Materials and Methods

### 2.1. Simulation of the PUMA 560 First Robot Axis

The PUMA 560 robot has DC motors with permanent magnets which was a common practice in the beginning of the robotic era. There are a lot of advantages to their use and favorable ratio between motor torque and velocity is certainly a major one. The reason behind it is the interaction between stator and rotor magnetic fields. This type of DC motors requires no energy for the stator. A consequence of that is less weight and volume for the same output power. Brushed permanent magnet DC motor, as its name says, has brushes used for transfer of DC electricity to the system. This motor generates torque directly from DC voltage using internal commutation, static permanent magnets and rotating electric magnets. Torque is generated by force (Lorentz force) at the ends of coil, positioned in an outside magnetic field. Motor contains internal inductance and resistance which can be approximated with an RL circle.

For the purpose of simulating the motion of the first robot axis it is necessary to take into consideration the mass of the whole robot, because the DC motor moves the complete structure, unlike other motors in the robot. For example, the last axis motor only operates with mass of the last segment of the robot. In Table 1, are presented all parameters of the first segment of the robot and mass for the whole robot.

For simulation purposes Robotics Toolbox for MATLAB/Simulink was used, developed by professor Peter Corke [3,16]. It is important to note that some parameters presented in Table 1 may vary for some robots. In few decades of production of robot PUMA 560, robots from different manufacturers emerged in the market and difference between them was mostly in the mass of some segments which has to be taken in consideration during simulation process [17]. For that reason, all parameters from [15] are experimentally verified for used robot and only those parameters are presented in Table 1. Mathematical model of robot PUMA 560 is described in detail in [15–17].

**Table 1.** Parameters of the PUMA 560 robot.

| Joint | Parameter | Value |
|-------|-----------|-------|
| 1st joint | Gear ratio Kr | 62.61 |
| | Encoder | 1000 imp/rev |
| | Accuracy | 0.101 mrad |
| | Length in Home position | 0.43 m |
| - | $J_{m,1}$ | $2 \times 10^{-4}$ kgm$^2$ |
| | $B_{m,1}$ | 6.3 Nms/rad |
| | $R_{a,1}$ | 2.1 $\Omega$ |
| | $K_{m,1}$ | 0.223 Nm/A |
| | $r_1$ | 62.61 |
| | $K_{b,1}$ | 0.26 V/rads |
| | $K_p$ | 260 |
| | $K_d$ | 80 |
| All joints | PUMA 560 mass | 54.5 kg |
| | Workspace | 320° |

Proportional plus derivative (PD) controllers are usually implemented independently at each joint of the robot. Assuming that the electric time constant is much smaller than the mechanical time constant, the dynamics of the j-th actuator of PUMA 560 robot (permanent magnet DC motor) can be presented in the following form:

$$\left[J_{m,j} + r_j^2 \cdot J_{r,j}(q)\right]\ddot{\theta}_{m,j} + \left(B_{m,j} + \frac{K_{b,j} \cdot K_{m,j}}{R_{a,j}}\right)\dot{\theta}_{m,j} = \frac{K_{m,j}}{R_{a,j}} \, v_{a,j} - r_j \cdot \tau_{r,j}(\theta) \tag{1}$$

where $\theta_{m,j}$ is the actuator angular position, $r_j$ is the gear reduction, $J_{m,j}$ is the sum of actuator and gear inertias, $B_{m,j}$ is the equivalent mechanical damping constant at the actuator shaft, $K_{m,j}$ is the motor torque constant, $R_{a,j}$ is the armature resistance, $r_j \cdot \tau_{r,j}(\theta)$ is the external load torque acting on the actuator axis, $v_{a,j}$ is the armature voltage, $K_{b,j}$ is the back emf constant of the actuator, $r_j^2 \cdot J_{r,j(q)}$ is the robot inertia reflected on the actuator shaft.

On the basis of the provided equation it is easy to draw a conclusion that actuator dynamics are linear in actuator angular position $\theta_{m,j}$, so therefore linear control theory can be applied. The PD controller is described by:

$$v_{a,j} = K_{p,j} \cdot \left(\theta_{m,j} - \theta_{m,j,d}\right) - K_{d,j} \cdot \dot{\theta}_{m,j} \tag{2}$$

where $\theta_{m,j,d}$ is the desired angular position and $K_{p,j}$ and $K_{d,j}$ are the proportional and derivative gains.

The closed loop system is obtained by applying the PD control to the actuator dynamics, which yields the following equation:

$$\left[J_{m,j} + r_j^2 \cdot J_{r,j}(q)\right]\ddot{\theta}_{m,j} + \left(B_{m,j} + \frac{K_{b,j} \cdot K_{m,j}}{R_{a,j}} + \frac{K_{m,j} \cdot K_{d,j}}{R_{a,j}}\right)\dot{\theta}_{m,j} + \frac{K_{m,j}}{R_{a,j}}K_{p,j} \cdot \left(\theta_{m,j} - \theta_{m,j,d}\right) + r_j \cdot \tau_{r,j}(\theta) = 0 \tag{3}$$

Equations (1)–(3) can be represented as a block diagram in Figure 1, created in Simulink:

Nonlinear and coupling effects of robot structure can be reduced by the mechanical gear reduction $r_j$. Taking into account the large gear reduction $r_j$, the change in magnitude of the robot configuration dependent terms $r_j^2 \cdot J_{r,j}(q)$ and $r_j \cdot \tau_{r,j}(\theta)$ may be neglected. Under such an assumption, they can be perceived as constant, linearising the system and allowing us to apply the Laplace transform to perform stability analysis. This assumption, however, will be revisited shortly.

Assuming that the $r_j^2 \cdot J_{r,j}(q)$ and $r_j \cdot \tau_{r,j}(\theta)$ are constant and applying Laplace transforms on Equation (3) we obtain the following:

$$\left\{\left[J_{m,j} + r_j^2 \cdot J_{r,j}(q)\right]s^2 + \left(B_{m,j} + \frac{K_{b,j} \cdot K_{m,j}}{R_{a,j}} + \frac{K_{m,j} \cdot K_{d,j}}{R_{a,j}}\right)s + \frac{K_{m,j}}{R_{a,j}}K_{p,j}\right\}\theta_{m,j}(s) = \frac{K_{m,j}}{R_{a,j}}K_{p,j} \cdot \theta_{m,j,d}(s) - r_j \cdot \tau_{r,j}(\theta) \tag{4}$$

The characteristic polynomial of the closed loop system is provided by

$$D(s) = \left[J_{m,j} + r_j^2 \cdot J_{r,j}(q)\right]s^2 + \left(B_{m,j} + \frac{K_{b,j} \cdot K_{m,j}}{R_{a,j}} + \frac{K_{m,j} \cdot K_{d,j}}{R_{a,j}}\right)s + \frac{K_{m,j}}{R_{a,j}}K_{p,j} \qquad (5)$$

from which we learn that any positive $K_{p,j}$, and $K_{d,j}$, will yield a stable closed loop system.

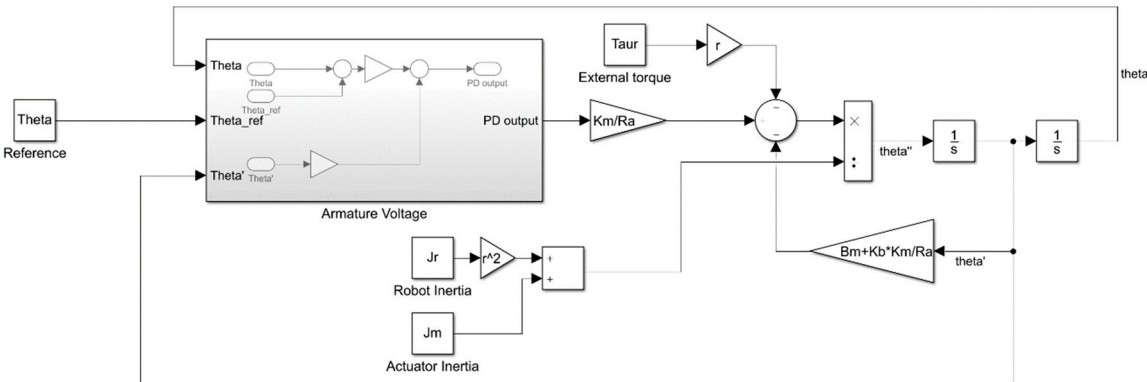

**Figure 1.** Basic control loop with plus derivative (PD) controller.

However, $r_j^2 \cdot J_{r,j}(q)$ and $r_j \cdot \tau_{r,j}(\theta)$ in a practical setting are configuration dependent terms, the zeros will drift on the complex plane as the robot moves. Therefore, it is difficult to obtain satisfactory performance of the closed loop system for all robot configurations if PD gains are fixed. Consequently, a tracking error occurs. The tracking error can only be reduced by using sufficiently large gains. Although a PD controller is very simple and robust, it suffers from the curse of "high gain". One way of achieving zero tracking error without using infinitely high gains is to compensate an external torque $r_j \cdot \tau_{r,j}(\theta)$ in Equation (4).

In Reference [15,18], it is proved that a PD controller with exact gravity compensation is asymptotically stable at the zero-equilibrium point for point-to-point control. In the matrix form Equation (1) can be written in the following form:

$$M(q)\ddot{q} + N(q,\dot{q})\dot{q} + g(q) = u \qquad (6)$$

where $q$ is the position vector of the joints of the robot, $u$ is the input torque or force acting on the joints, $M$ is the inertia matrix, $N$ is a matrix representing nonlinear centrifugal and Coriolis forces, and $g$ denotes the gravitational effect. On the assumption that the robot system (6) is controlled with the control law as presented in Figure 2:

$$u = g(q) + K_p \cdot (q_d - q) - K_d \cdot \dot{q} \qquad (7)$$

where $K_p$ and $K_d$ are two positive-definite gain matrices, the closed loop system is obtained as:

$$M(q)\ddot{q} = -N(q,\dot{q}) + K_p \cdot (q_d - q) - K_d \cdot \dot{q} \qquad (8)$$

Comparing Equations (6) and (7) with Equation (1) and doing term matching shows that we are compensating for the effects of terms which are a function of $q$, or equivalently, the gravitational effects of the manipulator configuration. Now, we proceed with the Lyapunov stability analysis (with nonlinearities in (6), we cannot use linear theory anymore), defining the position error as:

$$e = q_d - q \qquad (9)$$

and a Lyapunov function candidate:

$$V(q,\dot{q}) = \frac{1}{2}\dot{q}^T M(q)\dot{q} + \frac{1}{2}e^T K_p e \tag{10}$$

Differentiation of *V* along the closed loop system dynamics yields:

$$\dot{V}(q,\dot{q}) = \dot{q}^T M(q)\ddot{q} + \frac{1}{2}\dot{q}^T \dot{M}(q)\dot{q} - e^T K_p \dot{q} \tag{11}$$

$$\dot{V}(q,\dot{q}) = \dot{q}^T \cdot \left[u - g(q) - N(q,\dot{q})\dot{q}\right] + \frac{1}{2}\dot{q}^T \dot{M}(q)\dot{q} - e^T K_p \dot{q} \tag{12}$$

with Reference [13], $\dot{M}(q) - 2N(q,\dot{q}) = 0$ the following equation is:

$$\dot{V}(q,\dot{q}) = \dot{q}^T \cdot \left[-K_d \cdot \dot{q} - N(q,\dot{q})\dot{q}\right] + \frac{1}{2}\dot{q}^T \dot{M}(q)\dot{q} = -\dot{q}^T K_d \cdot \dot{q} \tag{13}$$

Equation (13) is negative for $\dot{q} \neq 0$. Therefore, $\dot{q}$ will reduce in magnitude until $\dot{q} \equiv 0$ which implies that $\ddot{q} = 0$. In this case, the closed loop system (8) yields e = 0.

At this point, it is worth noting that all assumptions made in this derivation are from a common set of assumptions for manipulators [15]. Their appropriateness has been confirmed in this work both by simulation and experiment.

The simulation result for the PUMA 560 robot with parameters from Table 1 is described in detail in [19], where the task was simulation of rotating first three axes in order to define appropriate controlling algorithm. Outdated robot construction with simple gear box solution dictated the use of the proposed control algorithm—a simple PD with gravity compensation shown in Figure 2. The aforementioned control algorithm can fully satisfy maximal precision which can be obtained from the PUMA 560 robot. On the other hand, proposed PD control algorithm required less resources, namely fewer multiplication units, which is important during its implementation on embedded platform. Figure 2 represents the control scheme in its traditional form [15]; the dynamics of the manipulator from (6) cannot be represented in such a scheme without resorting to a single block representation.

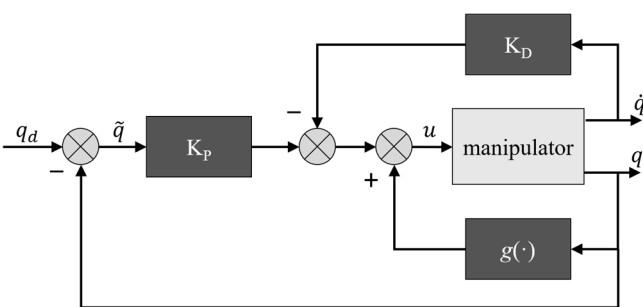

**Figure 2.** PD control with gravity compensation.

The proposed model was experimentally tested for three axes in [19], and it provides simulation for all three axes.

For the purpose of comparison, Equation (7) can be simplified using first robot axis because it is not affected by gravity g(q). All further analysis and experiments in this paper will hence restrict to the first axis. In Figure 3, the task was rotating the first segment of the robot from home position to 1 rad. Simulation result is presented in Figure 3.

A trajectory generated with polynomial of seventh degree [16,20] was used for simulation purposes. In the relevant literature it is also called the S trajectory which refers to the fact that it has 'slow' changing value of robot segment position during starting and stopping movements. Even the derivative of acceleration (known as jerk) is smooth function. Using the trajectory with S shapes causes

minimum stress to the robot's motor as well as other mechanical parts [16,20]. Position error obtained in simulation was around 1 mrad, but in the stationary state after movement it was significantly lower at 0.2 mrad. From the results obtained by simulation, it can be concluded that based on robot construction and age, simple control strategy can provide maximal precision in case of the PUMA 560 robot.

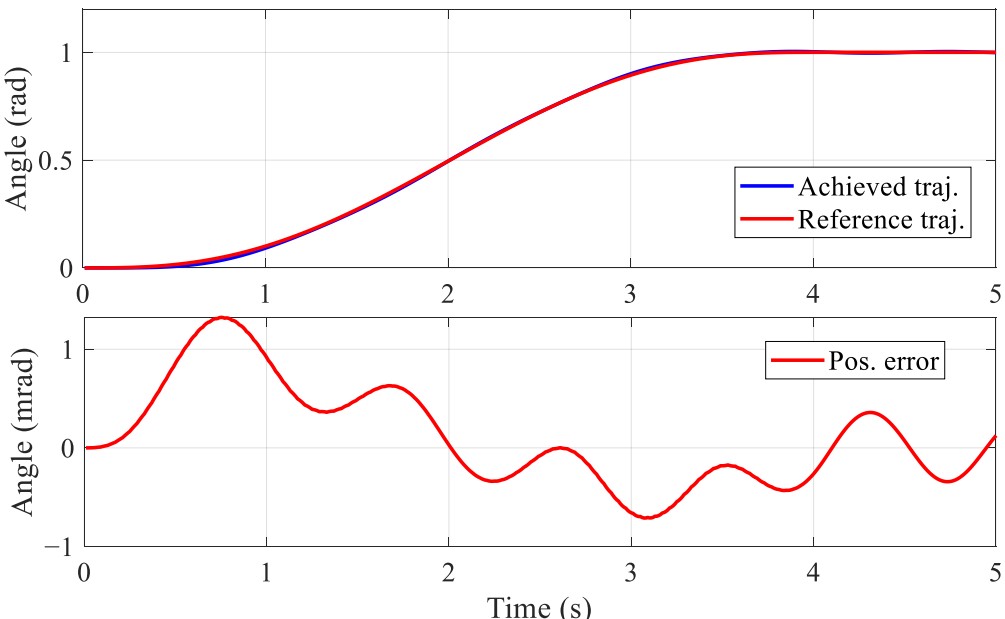

**Figure 3.** Simulation result for the first axis [19].

## 2.2. Control Scheme Using PC

The first controller model we present in this paper is a PC control scheme. Convenient for environments that already have general-purpose hardware and experience in usual engineering software, a PC scheme has an easy learning curve and allows fast implementation. Our inspiration here came from an educational setting: students gain experience in simulating devices represented by simplified transfer function models in a few hours—with just a few hours more, they can control the real system using the same software modality.

When using PC in real-time controlling applications, it is necessary to have appropriate hardware which allows connection between PC with installed control software (e.g., MATLAB) and the controlled object. In our case, Humusoft 634 data acquisition board was used for generating and accepting signals to PC from controlled object. It was placed inside a PC with 8 GB of RAM and i7 processor. The board has support from MATLAB and therefore signals from that environment can be taken into the MATLAB model and taken from model into the environment through analog and digital inputs/outputs. After installing the board into desktop PC, it is necessary to adjust the settings in MATLAB related to defining input/output ranges in Volts and to set up to Simulink for Real Time Windows Target (RTWT). Real-Time Windows Target is a one-box solution which allows PC to achieve real-time performance.

Signals from the PC, i.e., the Humusoft 634 board, should be presented in a suitable form to the robot. For that purpose, it was necessary to develop a driver with H bridge for controlling DC motors utilized in robots. The task of the driver is to translate low power signals from Humusoft board in the ±10 V range into a high-powered signal (60 V, ±10 A) for controlling DC motors. Figure 4 presents the driver we developed for this purpose, and it is described in detail in [21].

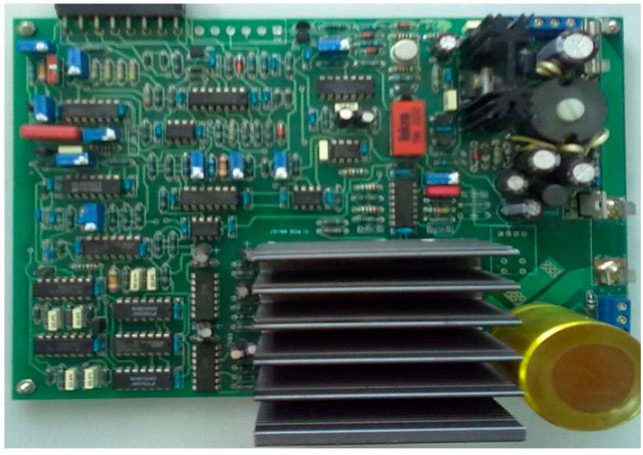

**Figure 4.** DC motor driver (60 V, ±10 A) [21].

The next task was acquiring the data from encoders attached to DC motors and providing data processing in order for them to be accepted into MATLAB Simulink environment. For that purpose, we developed a custom interface board, as presented in Figure 5 [22]. The interface board has interface circuitry for processing signals from older generation encoders which generate two 1 Vpp sine waves with the phase-shift of 90 degrees. It also provides DA and AD converters for connection of an FPGA, an RS-232 port, and optional connection to an acquisition board, in our case the Humusoft board.

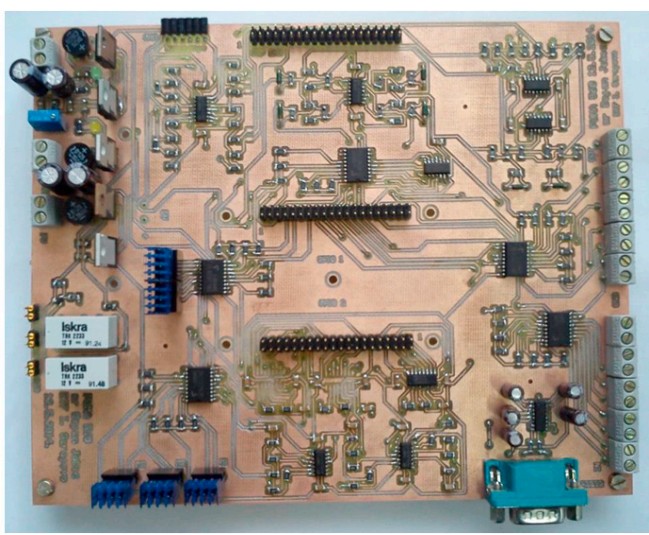

**Figure 5.** Interface board [22].

The interface board has many options and can be used for accepting the data from encoders in order to perform necessary processing and scaling to be accepted with Humusoft board.

It is important to note that it is possible to make closed control loop in MATLAB/Simulink using the information on the position of the DC motor shaft.

In Figure 6 we present the block scheme for PC-based control system of the PUMA 560 robot, with its three main parts.

MATLAB with Humusoft 634 data acquisition board which enables connection between the MATLAB/Simulink model and controlled objects was installed on desktop PC. In our case, the controlled object was the PUMA 560 robot which has DC motors for actuating a robot segment where the attached encoders on the shaft of each DC motor were used for measuring position. For the purpose of adjusting the signals from PC (Humusoft board) and robot (DC motors and encoders), a new controller was developed, and it is described in detail in [22]. Instead of an FPGA as in [22], the controller was driven

from the PC with Humusoft board. The above mentioned systems presented in block schemes enable testing modified PD regulator in the same manner as in the provided simulation.

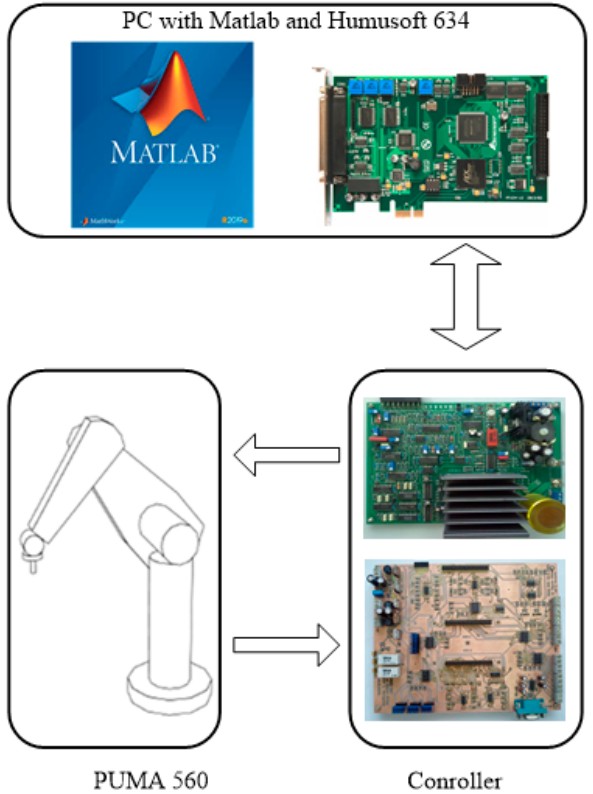

**Figure 6.** Block scheme for PC based control system of the PUMA 560 robot.

Figure 7 represents the Matlab Simulink software interface towards the manipulator, through the acquisition board. The 'dif' block corresponds to a discrete differentiator, and the rest of the scheme closes the loop with the controlled manipulator. The reference input is the generated seventh degree trajectory polynomial, generated in Robotics Toolbox and is used for all three cases, simulation, software control, and FPGA control. Replacement of the board and board-related software elements is straightforward (within input and output blocks), and in the best case, the board is replaced with an open architectural design as well. As such, the solution is ready for upgrades and modifications.

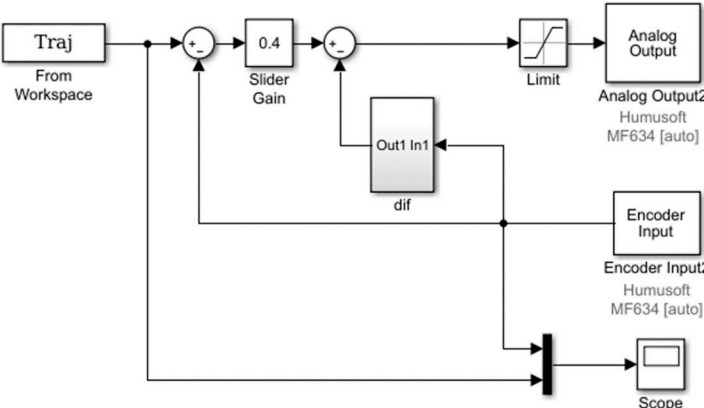

**Figure 7.** The Simulink controller using the acquisition board.

### 2.3. Control Scheme Using FPGA

An industrial setting, be it small or large, asks for dedicated hardware solutions for control, both for prototyping and for continuous use. Hence, our second scheme aims at allowing custom hardware development, but with as little restrictions caused by proprietary hardware/software as possible, while allowing an adjustable learning curve and budget trade-offs.

A controller based on an embedded portable system is in demand for many applications, so consequently it was decided to develop an FPGA-based controller. There are few different approaches: using microcontrollers, microprocessors, and digital signal processors (DSP) in combination, etc., but the most suitable approach is using FPGA due to its processing power and price [10–14]. Major drawback for using FPGA is programming using VHDL which is not adequate for developing control algorithms in user-friendly manner. For that purpose, it was decided to use MATLAB with the DSP Builder which enables design of control algorithms in a graphical environment. A major drawback for the DSP Builder is the absence of important blocks such as PD regulator, accepting signals from encoder etc. and for overcoming this problem new FPGA Real-Time Toolbox was developed, as described in detail in [23,24].

For the purpose of connecting the PC though FPGA with the PUMA 560 robot, we develop a modular structure akin to one in PC based control loop, represented in Figure 8 as a block scheme for the whole system.

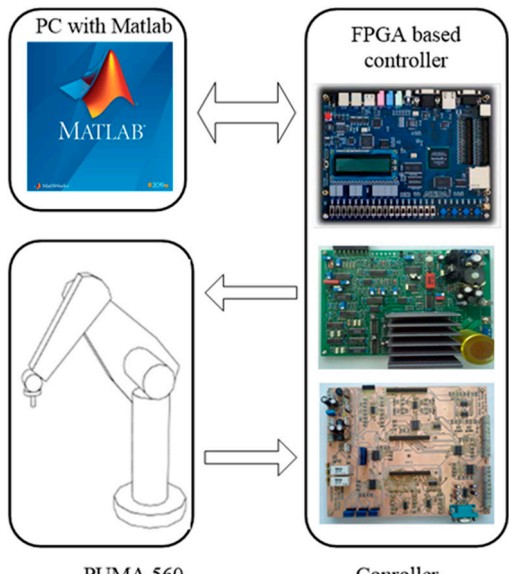

**Figure 8.** Block scheme of an field programmable gate arrays (FPGA)-based controller for the PUMA 560 robot.

In comparison with the PC-based controller, where a microprocessor on the desktop PC is used for calculating control algorithm in real time, in this case, we utilized the FPGA. The RS232 interface was used for collecting the data from FPGA in real time and the whole procedure is described in detail in [23].

## 3. Results

The experiment conducted in this paper had the goal of verifying the controllers we proposed, and the same task was put front of all three schemes: the simulation, PC, and FPGA control. They had to perform a rotational motion of the manipulator from 0 to 1 rad and back. Verification using a typical trajectory confirming that the error bounds are within the expected margin is a standard procedure for new open architectures [25], as the goal is to provide evidence for the applicability of their particular

implementations; the suitability of the underlying control-theoretic algorithm has already been shown for a variety of settings and manipulators [26].

Results of the PC control are presented in Figure 9. The first plot represents how the first robot axis follows the reference trajectory from 0 (home position) to 1 rad and returns to home position. It is important to note that for the first robot axis the load is maximal, because all the other robot segments of are attached to the first segment and total mass is 54.5 kg. Observing the results obtained for the first robot axis, it is impossible to notice any discrepancy between reference and achieved trajectory. For that reason, the second plot was provided, and it was obtained as an output signal from block error detection in the PD regulator. All values were obtained in number of pulses from encoders: 9850 pulses corresponds to 1 rad, i.e., a pulse corresponds to 0.102 mrad. Maximum measured error was 40 mrad and it is a consequence of the nature of PD regulator. Important results were obtained in the stationary state where the first axis rotating is finished, and its error is 2 pulses which equals 0.2 mrad, and is almost at the limit of encoder precision. Errors we cite here correspond to the worst case scenario of the robot starting up from prolonged inactivity (effect caused by the lubricant used for the robot's bearings). In all other scenarios, i.e., in motion of the robot after "warming up" measured errors were lower.

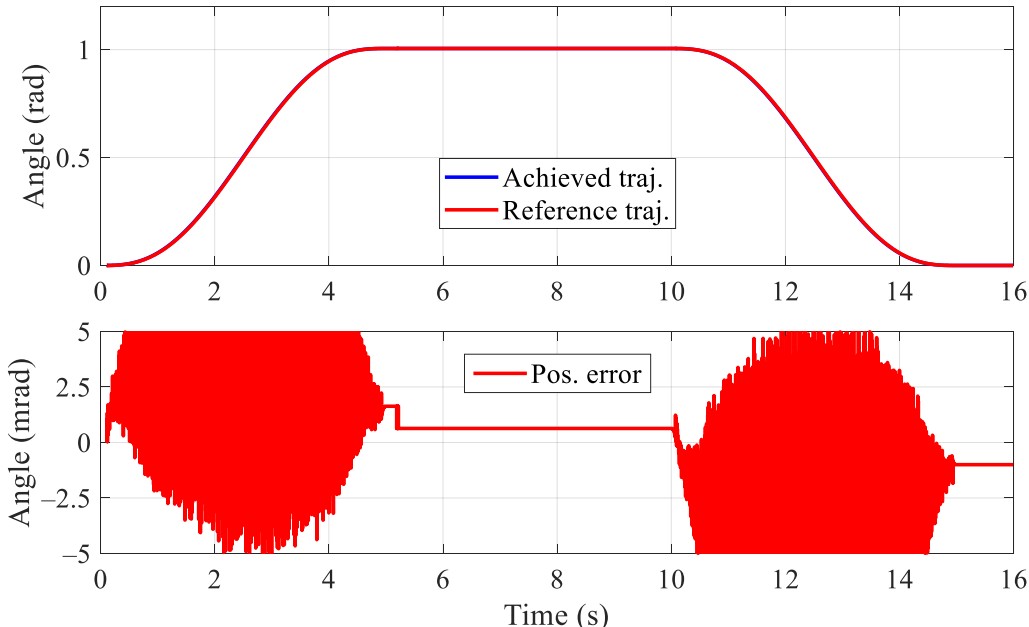

**Figure 9.** Experimental results for the first robot axis with PC control.

In Figure 10, we present the experimental results obtained using the FPGA-based controller. As in the previous experiment, the task was rotating the first axis from home position to the angle of 1 rad. During the experiment [19], the value of the worst case position error was obtained, and it was in the range of 10 to 12 pulses from encoders in the stationary state (the case in Figure 10 is not the worst, and it is comparable to the results from Figure 9). The error was significantly greater before reaching the stationary state which was expected due to the nature of the PD controller operating and the robot's mass of 54.5 kg (Table 1). The value of measured error before stationary state was out of range of the used measurement components (±2 mrad), and so it is not visible in the figures, but estimated indirectly. The worst-case scenario has the maximum measured error of 12 pulses in the stationary state, X = 0.0012 rad. The noise visible in Figure 9 (PC control) does not appear in Figure 10 (FPGA control) because of the AD/DA conversion: the Humusoft board used 14/bit converters, while in the FPGA scheme we used 8-bit converters.

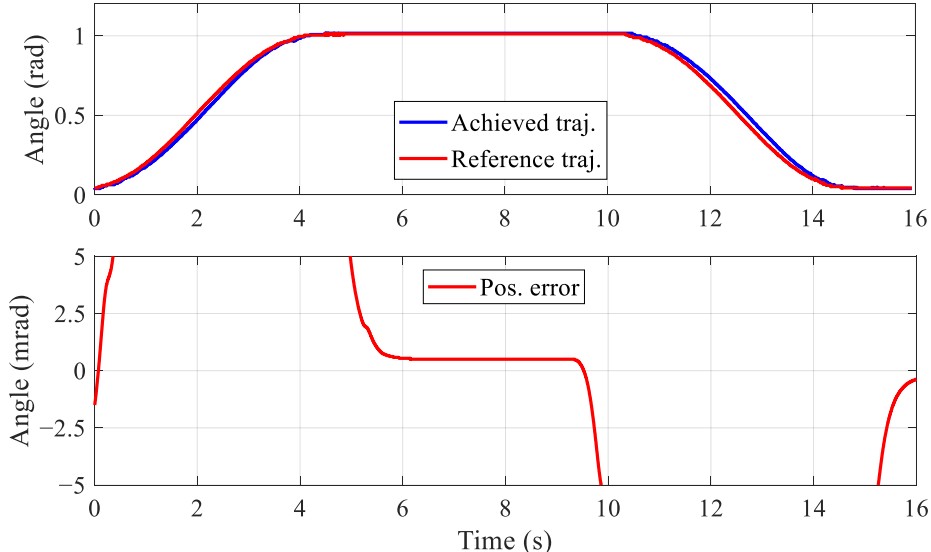

**Figure 10.** Experimental results for the first robot axis with FPGA control [19].

## 4. Discussion and Conclusions

In this paper, experimental results obtained using two different approaches are presented. The first approach is a PC-based controller with HUMUSOFT board which enables using Matlab/Simulink in real time. If we take into consideration advantages and disadvantages related to using PC in real-time applications and its reliability and vulnerability to viruses, this approach is education oriented. Nowadays, Matlab is common tool in education due to the fact that students are provided with the possibility to see any value from controlling algorithm. In that manner, a PC-based controller presents open architecture, which is an excellent tool in education process because students can compare signals from controller with signals from simulations. On the other hand, an FPGA-based controller is more industrial approach which can provide more reliable solutions. Programming the FPGA is very demanding but there are some benefits of using them in industrial applications such as parallel processing and dedicated logic for each task which enables reliability, efficiency and application specific integrated circuit (ASIC) solutions. When we speak of ASIC in this context, what we have in mind is the ability to generate a custom ASIC running our algorithm, eliminating both the PC and the FPGA from the loop, which is convenient for, e.g., mass production and industry integration. It is important to note that HDL can be generated from Matlab for the purpose of programing FPGA, but there are necessary steps to be implemented prior to downloading on FPGA. Complexity of programming the FPGA can be partially reduced using the DSP Builder which enables programming Altera's FPGA directly from Matlab. For that purpose is used, a self-developed FPGA Real-Time Toolbox [23]. In Table 2 we list features of both approaches.

**Table 2.** Features of PC and FPGA based controllers.

| Features (Low, Medium, High) | Type of Controller | |
|---|---|---|
| | **PC-Based** | **FPGA-Based** |
| Reliability | Low | High |
| Vulnerable on virus | High | Low |
| Skills | Low | High (Expert) |
| Industrial oriented | Low | High |
| Educational oriented | High | Low |
| Tuning parameters | High | Medium |
| Price of hardware | Medium | Low |
| Price of software | High | Low |
| Possibility of generating an ASIC deisgn | Low | High |

The use of Matlab in FPGA-based approach is optional, and all consequent customization of the HDL code can be done without it, hence the implication of low software cost in Table 2. In Reference [22], we have performed a cost analysis for PC+acquisition card vs. FPGA control solution, taking into account hardware costs and design labor costs, showing that the former requires ~35 thousand euros compared to 12 thousand euros for the latter. Without labor costs, i.e., just the hardware cost amounts for 15 thousand and 1 thousand euros, respectively.

Experiment we verified our controllers with was the rotational task of reaching 1 rad angle from the initial 0 rad state and going back to the initial state. The worst-case scenario error in the steady state was 0.2 mrad for the simulation and the PC-based controller, and 1.2 mrad for the FPGA based controller.

Identical results can be obtained from simulation and from experiment with PC-based controller (and the FPGA in the case of Figure 10), which is at the very edge of maximum possible accuracy which can be obtained with PUMA 560. It is important to note that the Robotics Toolbox with very precise model of the PUMA 560 was used for simulation purposes. These error margins confirm our approach is on par with state of the art control solutions for PUMA 560.

Significant discrepancy is noted between simulation results and the worst case results obtained using the FPGA-based controller. The reason is adjusting the control algorithm to the resources on FPGA, where number of multipliers is low, and number of bits used for representing numbers on FPGA structure must be reduced. Experimental results obtained using the PC-based controller indicated that error can be reduced using advanced FPGA in cases when it is unnecessary to reduce the number of bits in order to fit in the FPGA, which will be the topic of new further research (as listed in [19], the FPGA implementation for all three axes uses around 2250 logic cells, around 1000 dedicated logic registers, 70 DSP elements, 35 multipliers $18 \times 18$, 44 pins). An important question, of course, is that of the particular application—the FPGA based solution allows great simplification and scaling down of resources if the required accuracy is not high, as demonstrated in our example.

It is important to note that industrial FPGA-based solutions enable two important features: development of ASIC solutions and independence from vendors. Vendor independence is often the only guarantee of prolonged use of equipment, as for older types of robot vendors usually provide no support for maintenance or in case of failure, and the robot vendors become controller vendors. While software can sometimes allow independence, open hardware always allows it. Open software, hardware and their joint platforms are a stepping stone towards open primitives, open mechanical structures and an overall new paradigm of robotics.

The aim of this paper was to investigate different approaches, educational and industrial, for developing controllers for robots. For testing purposes, the PUMA 560 robot was used. From experimental and simulation results presented in the paper we may conclude that different approaches have their advantages and disadvantages, so we proposed the PC-based controller for educational and FPGA-based controller for industrial purposes. Experimental results also show that chosen hardware and self-developed hardware are adequate in order to meet necessary requirements for robot controllers, and the architecture allows scaling and improvements.

Making solutions like ours, practically verified, well-documented and easily customizable, available to the wide audience is not a contribution just to robotics and/or education audience: it is a contribution to circular economy, right to repair, and the innovation prospects of the community we live in. Robots discarded from production lines because of controller damage or discontinued support could be repurposed and allow enterprises that could not afford them otherwise to creatively use them in their manufacturing processes. Freedom of customization opens possibilities for variations: cooperative settings for multiple manipulators [26] or mobile manipulators come to mind [27].

**Supplementary Materials:** The following are available online at http://www.mdpi.com/2079-9292/9/6/972/s1: hardware description, PC control code, PCB design for the interface.

**Author Contributions:** D.J. and S.L. conceived the idea; D.J. helped with programming and writing the original draft preparation; D.J. and S.L. made substantial contributions to conception, design, analysis, and experimental verification; V.R. and M.B. contributed to review and editing the final version; H.Š. edited the final version and led the revision process. All authors have read and agreed to the published version of the manuscript.

**Funding:** This research was funded by the Ministry of Education, Science and Technological Development of the Republic of Serbia within the project No. III 43008.

**Conflicts of Interest:** The authors declare no conflicts of interest. The funders had no role in the design of the study; in the collection, analyses, or interpretation of data; in the writing of the manuscript, or in the decision to publish the results.

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
