# Peer review of "Two Open Solutions for Industrial Robot Control: The Case of PUMA 560"

_electronics, doi:10.3390/electronics9060972_

Round 1
Reviewer 1 Report
The authors present some experimental results (accuracy of robotic arm positioning) using software-in-a-loop solution and FPGA controller.
The hardware and algorithms were published elsewhere. The code is supplied with the paper for free-use by the readers. The paper may be useful for education purposes, but it needs major modifications.
Remarks:
In general, I have a problem identifying a new content presented by this paper - please clearly describe the purpose of the paper and novelty.
The paper lacks a detailed explanation of the root cause of discrepancies between FPGA and uP-based solutions implementing the same algorithm.
- table 1. please present accuracy in rads since you're referring to rads in the whole text. Also in the summary / experimental results, please refer to the datasheet parameters of the robot.
- line 126 - please fix spacing between the points, or explain the values in a single paragraph.
- line 194 - you should be more specific here.
- line 248 - please briefly say what's on the interface board.
Figure 2 - why showing first axis if later is evaluated another one. Why not showing simulation and measurements for both axes?
- line 286 - it's hard to say it's a "new controller architecture"
- line 294 - that's too basic. It's clear that the FPGA typically operate as stand-alone device. If PC software was in the loop there's no point using an FPGA.
- line 303 - this explanation is too basic. Just state what's on the graph.
- line 306 - just state the conversion factor: ...urad/pulse (from the datasheet, etc.).
- line 307 - I don't see 40 mrad on the plots.
- Fig. 8 why the error is so noisy? You state that the maximum error is 40mrad which is 4 times greater than for the FPGA system, but the plots for the FPGA case (fig. 9) are nice and clear.
- You're estimating error by the means of one measurement? you should show the statistics of many experiments.
- line 310 - what do you mean by "widely accepted error margin in digital systems". You're almost reaching the limit of the encoder resolution.
Fig. 8 - quality is low.
- line 330 - paper 18 was cited in the same content before. I think it's not necessary to say the same again here.
- line 331 - I don't understand this sentence.
- Table 2: what does ASIC mean in the content of the analyzed solution. ASIC is application-specific integrated circuit in my understanding.
- Table 2: Price of software: "Low" may not be true if Matlab is used for programming and development of the FPGA system.
- Table 2: it would be beneficial if the range of prices was mentioned.
The summary is very basic and does not show anything new than general, basic knowledge of the systems engineering.
Table 3. The experiment is not defined well and the methodology is not explained at all.
- line 370 - you should be more specific here, what models, how many multipliers do you need, relate to the resources usage of the existing system, etc.
- line 372: independence from vendors? It's not always true. The software code can be also reusable. As soon as you optimize and use vendor-specific primitives (which is often the case), the reusability of the FPGA is limited.
- line 372 - you should be also more specific here. What do you need ASIC for?
- line 374 - what do you mean? Why is this sentence here? What does it relate to?
You refer to the papers previously published by you on the same topic and using the same robot. Some of the papers were published many years ago. You should clearly state what's new in this paper and work.
Author Response
Dear editor, respected reviewers,
Thank you for your valuable comments that have helped us considerably improve our manuscript. We hereby submit a revised version of the manuscript: the changes are marked in blue (we have avoided using the default track changes mode due to its inconsistency) and their relationship with reviewers’ comments is given in the table below.
We hope that the revised manuscript answers all issues raised.
Best regards,
The Authors
---
Response to Reviewer 1
|
Comment |
Reply |
|
The authors present some experimental results (accuracy of robotic arm positioning) using software-in-a-loop solution and FPGA controller. The hardware and algorithms were published elsewhere. The code is supplied with the paper for free-use by the readers. The paper may be useful for education purposes, but it needs major modifications. |
Thank you for your comments. We have performed major rewriting as per your request. |
|
In general, I have a problem identifying a new content presented by this paper - please clearly describe the purpose of the paper and novelty. |
We have changed the introductory part of the paper and the paper’s ending to strongly emphasise the contribution of the paper to the open control platform community, and consequently, to the wider industrial and research audience, with effects expected in innovation and economy alike. See lines 48-52, 67-76, 414-419 and 427-433. |
|
The paper lacks a detailed explanation of the root cause of discrepancies between FPGA and uP-based solutions implementing the same algorithm. |
The line 360-362 now clarifies the first difference in the implementation causing discrepancies in control results: different AD/DA converters used. Consequentially, the limited resources on the used FPGA are discussed in line 406-412. |
|
table 1. please present accuracy in rads since you're referring to rads in the whole text. Also in the summary / experimental results, please refer to the datasheet parameters of the robot. |
Table 1 has been updated with accuracy in rads (line 125). |
|
line 126 - please fix spacing between the points, or explain the values in a single paragraph. |
The values are now listed in a single paragraph (line 139-143). |
|
line 194 - you should be more specific here. |
We have specified what less resources means—fewer multiplication units. It is now in line 210. |
|
line 248 - please briefly say what's on the interface board. |
We have now introduced the components on the interface board in line 264-268. |
|
Figure 2 - why showing first axis if later is evaluated another one. Why not showing simulation and measurements for both axes? |
Both the simulation and the experiments refer only to the first axis in this paper. We now reiterate that in line 219-220, 223-224. |
|
line 286 - it's hard to say it's a "new controller architecture" |
We now call it a modular structure in line 318. |
|
line 294 - that's too basic. It's clear that the FPGA typically operate as stand-alone device. If PC software was in the loop there's no point using an FPGA. |
The superfluous explanation has been removed from line 324. |
|
line 303 - this explanation is too basic. Just state what's on the graph. |
The explanation has been removed, the line 335-336 now states what is on the graph. |
|
line 306 - just state the conversion factor: ...urad/pulse (from the datasheet, etc.). |
The conversion factor is now stated in line 337 |
|
line 307 - I don't see 40 mrad on the plots. |
As stated in line 358, the error in that part of the trajectory exceeds the range of the measurement components, and it was calculated indirectly in [18]. Line 359 was edited to clarify this. Also, with the update of Fig. 9, explanation in line 353-355 was updated to reflect what is visible in the figure. |
|
Fig. 8 why the error is so noisy? You state that the maximum error is 40mrad which is 4 times greater than for the FPGA system, but the plots for the FPGA case (fig. 9) are nice and clear. |
The line 360-362 now clarifies the cause of the noise: different AD/DA converters used. |
|
You're estimating error by the means of one measurement? you should show the statistics of many experiments. |
We now clarify that the error estimate is the worst case scenario in line 340-343. |
|
line 310 - what do you mean by "widely accepted error margin in digital systems". You're almost reaching the limit of the encoder resolution. |
The “widely accepted error” claim has been substituted with a comment about the limit of encoder precision in line 340. |
|
Fig. 8 - quality is low. |
Both Figs. 8 and 9 have been replaced with better quality graphs. (line 347 and 350) |
|
line 330 - paper 18 was cited in the same content before. I think it's not necessary to say the same again here. |
The elaboration and the citation were removed from line 360. |
|
line 331 - I don't understand this sentence. |
The sentence was reformulated in line 359-360. |
|
Table 2: what does ASIC mean in the content of the analyzed solution. ASIC is application-specific integrated circuit in my understanding. |
This has now been clarified both in the table 2 and in line 376-379 in the text. |
|
Price of software: "Low" may not be true if Matlab is used for programming and development of the FPGA system. |
We have now clarified in line 388-389 that Matlab dependency can be removed from this approach, keeping the price low. |
|
Table 2: it would be beneficial if the range of prices was mentioned. |
A range of prices has now been included in text (line 390-393) as a part of discussion, based on [23]. |
|
The summary is very basic and does not show anything new than general, basic knowledge of the systems engineering. |
The experiment itself serves only as verification of controller suitability. This has now been clarified in line 394. |
|
Table 3. The experiment is not defined well and the methodology is not explained at all. |
The methodology and the experimental results are now presented in textual form, line 394-412. |
|
line 370 - you should be more specific here, what models, how many multipliers do you need, relate to the resources usage of the existing system, etc. |
We have now updated line 408-410 with the information on the resources of the FPGA used in the controller design. |
|
line 372: independence from vendors? It's not always true. The software code can be also reusable. As soon as you optimize and use vendor-specific primitives (which is often the case), the reusability of the FPGA is limited. |
We have addressed this concern in line 414-419, suggesting our work as a part of a bigger picture in movement away from vendor dependence. |
|
line 372 - you should be also more specific here. What do you need ASIC for? |
This has now been addressed with the elaboration on ASIC in line 376-379. |
|
line 374 - what do you mean? Why is this sentence here? What does it relate to? |
This sentence was supposed to be an explanation for vendor independence argument. This has now been elaborated in line 414-419. |
|
You refer to the papers previously published by you on the same topic and using the same robot. Some of the papers were published many years ago. You should clearly state what's new in this paper and work. |
We have now included a paragraph, line 67-76 to explain the position of this work with respect to our earlier work. |

Reviewer 2 Report
This paper has minor contribution
It is evident that the authors have done a great job and, in its sector, extremely interesting. Unfortunately, the paper does not highlight the main innovation or the simple technological application in a sufficiently clear way: it seems that it is still "immature" and requires further work.
I suggest a complete rewrite, using other papers that deal with similar topics as a trace.
Some comments:
- The mathematical part is interesting and sufficiently correct but it is presented in a confusing way. For example, some assessments are not justified and, as a final algorithm, is presented only a simplified block diagram (Fig. 1) which must then be correlated with the Expression (13)
- the first part (software) is not described with sufficient clarity, moreover the part concerning the Matlab section is not described at all: it would have been interesting to see the management of the problem with Simulink. In any case, the innovative part is not sufficiently highlighted
- The second part (FPGA programming) is full of details on the tools that have been used but very vague in the essential details: here too, the innovative part is not highlighted
- the results are presented in an unclear and extremely confusing way: it could be useful to completely rewrite them, highlighting clearly and unequivocally what the results have been achieved, with the initial conditions, data, etc.
- The work also lacks a comparative part between the proposed methods and the state of the art of the controls of the PUMA 560
- Most of the conclusions are known facts.
- A moderate English review is required
Minor comments:
- Figures 3, 4 and 5 are not necessary: the boards have not been developed or customized by the authors. Mentioning their part number is more than enough to trace the components used
- Figure 8 has a resolution that makes it virtually useless. In the picture below we can only distinguish spots of colour: please increase the resolution and the size
- Figure 9 is not clear: please improve resolution
I also suggest integrating the References with at least these three papers:
- Leccese et al., "A Simple Takagi-Sugeno Fuzzy Modelling Case Study for an Underwater Glider Control System," 2018 IEEE International Workshop on Metrology for the Sea; Learning to Measure Sea Health Parameters (MetroSea), Bari, Italy, 2018, pp. 262-267, doi: 10.1109/MetroSea.2018.8657877
- Ikeda and M. Minami, "Asymptotic stable guidance control of PWS mobile manipulator and dynamical influence of slipping carrying object to stability," Proceedings 2003 IEEE/RSJ International Conference on Intelligent Robots and Systems (IROS 2003) (Cat. No.03CH37453), Las Vegas, NV, USA, 2003, pp. 2197-2202 vol.3, doi: 10.1109/IROS.2003.1249197.
- Petritoli, F. Leccese, L. Ciani and G. S. Spagnolo, "Probe Position Error Compensation in Near-field to Far-field Pattern Measurements," 2019 IEEE 5th International Workshop on Metrology for AeroSpace (MetroAeroSpace), Torino, Italy, 2019, pp. 214-217, doi: 10.1109/MetroAeroSpace.2019.8869674.
Author Response
Dear editor, respected reviewers,
Thank you for your valuable comments that have helped us considerably improve our manuscript. We hereby submit a revised version of the manuscript: the changes are marked in blue (we have avoided using the default track changes mode due to its inconsistency) and their relationship with reviewers’ comments is given in the table below.
We hope that the revised manuscript answers all issues raised.
Best regards,
The Authors
Response to Reviewer 2
|
Comment |
Reply |
|
This paper has minor contribution
It is evident that the authors have done a great job and, in its sector, extremely interesting. Unfortunately, the paper does not highlight the main innovation or the simple technological application in a sufficiently clear way: it seems that it is still "immature" and requires further work.
I suggest a complete rewrite, using other papers that deal with similar topics as a trace. |
Thank you for your comments. We have performed major rewriting as per your request. In our rewrite, which has followed similar papers contributing to the body of open architectures, we have focused on demonstrating our contribution. For example, the expansion of the introductory and the concluding part of the work, lines 48-52, 67-76, 414-419 and 427-433.highlight the motivation for this work and its place in the wider picture. |
|
The mathematical part is interesting and sufficiently correct but it is presented in a confusing way. For example, some assessments are not justified and, as a final algorithm, is presented only a simplified block diagram (Fig. 1) which must then be correlated with the Expression (13) |
The assumptions made in the derivation are standard, textbook assumptions, and verification of the algorithm by both simulation and experiment confirm their appropriateness. We reinforce this in line 200-202. We also explain why Fig. 1 is the only possible graphical representation of the control loop in line 211-213. |
|
the first part (software) is not described with sufficient clarity, moreover the part concerning the Matlab section is not described at all: it would have been interesting to see the management of the problem with Simulink. In any case, the innovative part is not sufficiently highlighted |
The paragraph in line 292-298 presents a new figure (Fig. 6) with the Matlab solution, and gives details on the implementation. |
|
The second part (FPGA programming) is full of details on the tools that have been used but very vague in the essential details: here too, the innovative part is not highlighted |
We have now updated line 406-412 with the information on the resources of the FPGA used in the controller design. |
|
the results are presented in an unclear and extremely confusing way: it could be useful to completely rewrite them, highlighting clearly and unequivocally what the results have been achieved, with the initial conditions, data, etc. |
The methodology and the experimental results are now presented in textual form, line 394-412. |
|
The work also lacks a comparative part between the proposed methods and the state of the art of the controls of the PUMA 560 |
With the exhibited error profiles, we can confirm that our method is on par with the state of the art, and we now state it in line 398-402. |
|
Most of the conclusions are known facts. |
We have expanded the conclusions so that the purpose of this paper is clear (line 427-433). |
|
A moderate English review is required |
We have proof-read the paper and fixed certain language issues. |
|
Figures 3, 4 and 5 are not necessary: the boards have not been developed or customized by the authors. Mentioning their part number is more than enough to trace the components used |
We have removed Fig. 3 as per your comments. Figures 4 and 5 (now Figs. 3 and 4) represent boards developed by the authors, and we have updated the text (lines 257-258 and 264-268) to clearly state that. |
|
Figure 8 has a resolution that makes it virtually useless. In the picture below we can only distinguish spots of colour: please increase the resolution and the size |
The Figure has been updated to a better resolution (line 347) |
|
Figure 9 is not clear: please improve resolution |
The Figure has been updated to a better resolution (line 349) |
|
I also suggest integrating the References with at least these three papers:
Leccese et al., "A Simple Takagi-Sugeno Fuzzy Modelling Case Study for an Underwater Glider Control System," 2018 IEEE International Workshop on Metrology for the Sea; Learning to Measure Sea Health Parameters (MetroSea), Bari, Italy, 2018, pp. 262-267, doi: 10.1109/MetroSea.2018.8657877 Ikeda and M. Minami, "Asymptotic stable guidance control of PWS mobile manipulator and dynamical influence of slipping carrying object to stability," Proceedings 2003 IEEE/RSJ International Conference on Intelligent Robots and Systems (IROS 2003) (Cat. No.03CH37453), Las Vegas, NV, USA, 2003, pp. 2197-2202 vol.3, doi: 10.1109/IROS.2003.1249197. Petritoli, F. Leccese, L. Ciani and G. S. Spagnolo, "Probe Position Error Compensation in Near-field to Far-field Pattern Measurements," 2019 IEEE 5th International Workshop on Metrology for AeroSpace (MetroAeroSpace), Torino, Italy, 2019, pp. 214-217, doi: 10.1109/MetroAeroSpace.2019.8869674. |
We have expanded the references with two additional papers, one of which is your proposal—the Ikeda & Minami paper (see context in line 432-433). The two other papers were a difficult match, as they are hard to link with robotic manipulators. |

Round 2
Reviewer 1 Report
Thank you for implementing most of my remarks. I recommend for publication.
Author Response
We were happy to read that you approve of the change made
Reviewer 2 Report
Unfortunately, despite a major revision, the paper has not been significantly improved.
I greatly appreciated the effort that the authors have made but it remains below the standard and must be rejected.
My comments on the paper remain roughly the same of the first revision: i.e: the mathematical part has not been modified except marginally, the results are still presented in a confusing way and the innovative part is not clear and still minor, etc...
I still suggest a complete rewrite, using other similar papers as a trace.
Author Response
Dear editor, respected reviewers,
Thank you for your valuable comments that have helped us once again to improve our manuscript. We hereby submit a revised version of the manuscript: the changes are marked in dark red (the previous round of revisions is marked in blue) and their relationship with reviewers’ comments is given below.
We hope that the revised manuscript answers all issues raised.
Best regards,
The Authors
---
Response to Reviewer 2
Unfortunately, despite a major revision, the paper has not been significantly improved.
I greatly appreciated the effort that the authors have made but it remains below the standard and must be rejected.
We do hope that this new revision brings significant improvements to your liking, as they focused on making the mathematical part easier to approach, results more meaningful, and the contribution more obvious.
My comments on the paper remain roughly the same of the first revision: i.e: the mathematical part has not been modified except marginally, the results are still presented in a confusing way and the innovative part is not clear and still minor, etc...
- We have gone back to a recent similar paper (reference [28] in our paper, Martínez-Prado, M.A., Rodríguez-Reséndiz, J., Gómez-Loenzo, R.A., Herrera-Ruiz, G. and Franco-Gasca, L.A., 2018. An FPGA-based open architecture industrial robot controller. IEEE Access, 6, pp.13407-13417) and made sure that our paper follows the same structure as theirs did. We emphasise this in our revised manuscript, line 360 in the context of the experimental setup and results discussion. When new open architecture solutions are introduced, the point of the experiment is to verify their functionality; a comparison with other methods is beyond the scope, as the control-theoretic algorithm is not under test here, but its implementation (we clarify that in line 366). References [28] and [29] were added for this purpose.
- To demonstrate the innovation potential of our work, we have added introductions to design description sections that explain how they fit in two use-cases we had in mind, education and industry (line 264, line 334).
- The abstract has been updated to show the necessity for our and similar work of others (line 17). A paragraph has been added to the introduction (line 52) to show the fundamental reasoning behind the urgency of providing open platforms to both industry and education, with two new references whose titles alone tell part of the story ([1], Shah, A. (September 1, 2018). "Can You Repair What You Own?." ASME. Mechanical Engineering. September 2018; 140(09): 37–41, [2] Mondada, F., Bonani, M., Riedo, F., Briod, M., Pereyre, L., Rétornaz, P. and Magnenat, S., 2017. Bringing robotics to formal education: The thymio open-source hardware robot. IEEE Robotics & Automation Magazine, 24(1), pp.77-85.)
- The mathematical part has been updated with a new figure (Fig. 1, line 170) which now represents the model for the differential equations (1-3). For the modelling community, such an illustration is helpful and hopefully it increases the readability. Reviewer’s concerns about assumptions have been addressed in line 174, suggesting that the external, configuration dependent terms are taken constant just in the zeroth approximation. In line 205, we clarify the link between equations (1) and (6) and justify the use of nonlinear tools in the remainder of the treatise.

Round 3
Reviewer 2 Report
The paper is accepted